# Evaluation of the Effect of Different Dietary Lipid Sources on Dogs’ Faecal Microbial Population and Activities

**DOI:** 10.3390/ani12111368

**Published:** 2022-05-26

**Authors:** Alessandro Vastolo, Jonathan Riedmüller, Monica Isabella Cutrignelli, Jürgen Zentek

**Affiliations:** 1Department of Veterinary Medicine and Animal Production, University of Napoli, Federico II, 80137 Napoli, Italy; alessandro.vastolo@unina.it; 2Institute of Animal Nutrition, Freie Universität Berlin, 141595 Berlin, Germany; johnyreed@zedat.fu-berlin.de (J.R.); juergen.zentek@fu-berlin.de (J.Z.)

**Keywords:** fatty acids profile, hemp, tallow, *Lactobacillus*, volatile fatty acids, in vitro fermentation, bacteria, inulin, faecal inoculum

## Abstract

**Simple Summary:**

Saturated fatty acids might be a valuable source of energy to guarantee all physiological functions in companion animals. Polyunsaturated fatty acids are essential in several metabolic processes and structural body functions. In this regard, hemp oil can be used as a rich source of polyunsaturated fatty acids in animal diets. In this study, hemp seed oil and swine tallow were added to a commercial canned diet. These high-lipid-content diets (hemp diet: 55.1 g/100 kcal ME; tallow diet: 65.1 g/1000 kcal ME) were compared with one rich in starch. Following the recruitment of 12 dogs, three experimental groups were set up. At 30 days of diet administration, faeces samples were collected from each group to perform an in vitro trial and faecal bacteria count. In the first evaluation, the faecal inoculum obtained from dogs fed a diet supplemented with hemp showed higher fermentation activity and lower gas production at 24 h of incubation. The bacterial count demonstrated an increase in *Lactobacillus* when hemp group faeces were tested. Both in vivo and in vitro acetic acid production increased. The results obtained suggest an influence of the fatty acid profile on the microbial population.

**Abstract:**

Lipids represent a significant energy source in dogs’ diets. Moreover, dogs need some essential fatty acids, such as linoleic and α-linolenic fatty acids, because they are not able to produce them endogenously. This study aimed to evaluate the effect of different dietary lipid sources on faecal microbial populations and activities using different evaluations. Hemp seed oil and swine tallow were tested as lipid supplements in a commercial canned diet at a ratio of 3.5% (HL1 and HL2, respectively). These diets were compared with one rich in starch (HS). Twelve dogs were recruited and equally divided into three groups. Faeces samples at 30 days were used as inoculum and incubated with three different substrates (MOS, inulin, and cellulose) using the in vitro gas production technique. The faecal cell numbers of relevant bacteria and secondary metabolites were analysed (in vivo trial). In vitro evaluation showed that the faeces of the group fed the diet with hemp supplementation had better fermentability despite lower gas production. The in vivo faecal bacterial count showed an increase in *Lactobacillus* spp. In the HL1 group. Moreover, a higher level of acetate was observed in both evaluations (in vitro and in vivo). These results seem to indicate a significant effect of the dietary fatty acid profile on the faecal microbial population.

## 1. Introduction

Carnivores tolerate lipids well, which represent an essential energy source thanks to their high digestibility [1]. In this regard, saturated fatty acids satisfy energy requirements in order to regulate body temperature, growth, reproduction, and voluntary physical activities in companion animals. Moreover, these nutrients might be an energy reserve source for future mobilization in case of necessity [2]. Consequently, it is necessary to balance the lipid content with other nutrients amount in order to satisfy all nutritional requirements. In this context, medium-chain triglycerides (MCTs) are saturated fatty acids with 6 to 12 carbon atoms that appear to be rapidly digested by lipases and transported directly into the bloodstream, as described in pigs by Zentek et al. [3]. These triglycerides have been suggested for companion animals with lipid digestion problems and poor physical condition. Furthermore, Han et al. [4] recently indicated medium-chain triglycerides as an auxiliary treatment for humans, rodents, and dogs with epilepsy. Indeed, these triglycerides provide additional energy in the central nervous system by compensating for the lack of energy due to current antiepileptic drugs. 

Dogs are not able to produce n-6 linoleic acid (LA) and n-3 α-linolenic (ALA) fatty acids endogenously. Therefore, these fatty acids are essential (EFA) in canine diets [5]. The n-3 and n-6 families represent several fatty acids derived from LA and ALA. These groups of essential fatty acids are involved in several metabolic processes and structural body functions as fundamental components of cell membranes that participate in the transport of nutrients and metabolites across membranes. Furthermore, they are necessary for regular growth and prevention of several health disorders (e.g., cardiovascular diseases, diabetes, hypertension, chronic inflammatory, autoimmune disorders, and cancer) in mammalians [6]. Most plant oils contain between 80 and 90% of unsaturated fatty acids, while animal fats have between 50 and 60% of unsaturated fatty acids [7]. In this regard, canola, corn, soybean, and sunflower oil represent vegetable lipid sources more used in the pet food industry [5]. 

Hemp oil has shown beneficial effects thanks to its high polyunsaturated essential fatty acids content (over 80% in hemp seed oil), and the presence of minor compounds such as ß-sitosterol, campesterol, phytol, cycloartenol, and tocopherol [8]. It contains linoleic acid and α-linolenic acid as its major n-6 and n-3 polyunsaturated fatty acid (PUFA) [9]. Moreover, some authors have demonstrated that hemp oil might be a rich source of polyunsaturated fatty acids in feed mixtures for animals. In addition, hemp seed and hempseed cakes represent fat and protein sources for farm animals [10,11,12]. On this matter, the residue of hemp (*Cannabis sativa* L.) oil processing could be a valid resource for lipids and essential fatty acids (LA and ALA) in dogs’ diets [13]. However, despite the authorization by the European Food Safety Authority (EFSA) to use hemp seed and hempseed co-products as ingredients for all animal species [14], few data have been published in the literature concerning the use of hemp co-products as a source of PUFAs in pet food. 

The gut microbial population in canine species has shown high individual variability [15,16]. Various factors affect the microbial population in the gut, such as age, environment, and dietary composition [17]. In this respect, nutrition represents one of the most important factors able to affect the composition and metabolism of microbiota. In the last years, researchers have evaluated the effect of different nutrients [18] or physical forms of diet [15,16,17,18,19] on the microbial population. Numerous studies described the positive effect of carbohydrate sources on the microbiota composition. However, there is a lack of data regarding the effect of the lipid quantity and quality on dogs’ faecal microbial population [15,16,17,18,19]. Recently, Kilburn et al. [20] investigated the effect of different canola oil inclusions (32, 37, 42, 47 of % fat) in the diet of adult Beagles on the faecal microbiome. The authors did not observe a negative effect on the microbial community and health of the animals.

This study aimed to evaluate the effect of different dietary lipids on faecal microbial populations and activities. For this purpose, two different supplementations of lipid sources: one of animal origin (swine tallow) and the other of vegetable origin (hemp seed oil), were tested. 

## 2. Materials and Methods

### 2.1. Animal and Diets

In total, 12 crossbreed neutered dogs (age 6 ± 1.65 years; weight 19.13 ± 5.04 kg; body condition score BCS 4.75 ± 0.87 on a 9-point scale) recognized as healthy by clinical evaluation and haematological, biochemical, and parasitological tests were recruited from a municipal kennel located in Naples. Three groups homogeneous for sex, body weight, and BCS were created. Each dog was housed in an individual box of 12 m^2^ (3 × 4 m^2^) divided between a closed rest section (1 × 2 m^2^) and an open area. The experiment lasted for 40 days (10 days of diet adaptation and 30 days of diet administration). 

A double-blind nutritional trial was performed by supplementing a canned commercial diet with the same amount (3.5% as fed) of two lipid sources (swine tallow or hemp seed oil). The latter was the residue of hemp seed oil extraction, which was unfiltered during the processing. The obtained diets, named High Lipid 1 (HL1) and High Lipid 2 (HL2), were compared with a canned diet that was richer in starch (HS) and usually utilised in the kennel. The latter diet was considered the control diet. Each diet was randomly assigned to one group and administered daily at a ratio of 110 kcal/kg^0.75^ of metabolizable energy (ME) [21].

Table 1 gives an overview of the chemical composition of the diets. Each diet was analysed in triplicate regarding its dry matter (DM), crude protein (CP), ether extract (EE), crude fibre (CF), and ash content according to AOAC [22] procedures (ID number: 2001.12, 978.04, 920.39, 978.10, 930.05, and 996.11, respectively). The nitrogen-free extracts (NFEs) were calculated by the following equation: NFE= 100 − (CP + CF + EE + ash). The metabolizable energy density was calculated using the modified Atwater factors. Each diet was replicated two times for the fatty acids profile (Table 2), which was performed after total fat extraction [23]. According to Christie [24], each sample was turned into methyl esters by direct transesterification. The fatty acids profile was determined using a gas chromatographic system (Focus GC: Thermo Electron Corporation, Waltham, MA, USA) with a flame ionization detector (FID) for methyl esters [25]. An external standard mixture (Larodan Fine Chemicals: AB, Limhamnsgardens, Malmo, Sweden) was used to quantify each fatty acid amount.

### 2.2. In Vitro Fermentation

After 30 days of diet administration, fresh faeces were collected. Faecal samples were immediately transported in thermostatic boxes under anaerobic conditions to the laboratory of the Feed Evaluation of the University of Napoli, Federico II. Individual samples of each group were pooled in order to perform the in vitro gas production technique [26]. Each pool of faeces was diluted (1:10 *v*:*v*) with 0.9% NaCl sterile solution, homogenized and filtered through a cheese cloth, and added to each flask (5 mL). Then, the flasks were incubated at 39 °C for 24 h [27]. Three carbohydrates sources (cellulose, inulin, and mannan oligosaccharides), differing in fermentation characteristics, were chosen as substrates. Four replications for each pool of faeces and each substrate were incubated, including four blanks (flask with *inoculum* without substrates). The gas production of the fermenting cultures was recorded with a manual pressure transducer (Cole and Parmer Instrument, Vernon Hills, IL, USA) during the incubation. The cumulative volume of gas obtained for each sample at 24 h was related to the quantity of incubated organic matter (OMCV, mL/g). The fermentation was stopped by cooling the flasks at 4 °C. Two aliquots (5 mL) of fermenting liquor were collected and frozen at −15 °C for volatile fatty acids (VFAs) analysis, and the pH of the fermenting liquor was measured with a pHmeter (ThermoOrion 720 A+, Fort Collins, CO, USA). The volatile fatty acids were determined using gas chromatography (ThermoQuest Italia SpA, Rodano, Milan, Italy; model. Focus) as indicated by Calabrò et al. [28]. To evaluate the degree of proteolysis that occurs during fermentation, the branched-chain fatty acids were calculated according to the following equation: BCFAs = (Iso-butyrate + Iso-valerate)/VFAs. The gas profile of each flask was fitted to the sigmoid model described by Groot et al. [29] as follows: (1)$$G=A/(1+{\left(B/t\right)}^{c})$$
where G is the total gas produced (mL/g of incubated OM) at t (h) time; A is the asymptotic gas production (mL/g of incubated OM); B is the time at which one-half of the asymptote is reached (h); and C is the switching characteristic of the curve.

### 2.3. Bacterial Cell Evaluation

Individual faecal samples were collected to analyse the bacterial cell numbers, D-L-lactate, and ammonia at 30 days of diet administration. All analyses were performed at the Institute of Animal Nutrition, Freie Universität Berlin. The DNA was extracted from dogs’ faeces (250 mg) using a commercial kit (QiaAmp Stool Kit: Qiagen, Hilden, Germany) as described by Kröger et al. [30]. Successively to fluorescent identification of the DNA amount, the extracts were corrected to 20 ng DNA/μL (NanoDrop 2000: Thermo Fisher Scientific, Darmstadt, Germany). Additionally, a commercial master mix (BrilliantSYBRgreen II: Stratagene, Amsterdam, Netherlands) was used together with the primers given in Table 3 in order to amplify the samples in a Real-Time PCR Cycler (MX3005P: Stratagene, Amsterdam, The Netherlands).

The assessment of volatile fatty acids (VFAs) was performed as described in the previous paragraph. The D- and L-lactate faecal amounts were determined using HPLC (Agilent 1100: Agilent Technologies, Böblingen, Germany) with a pre-column (Phenomenex C18 4.0 4.0 × 2.0 mm: Phenomenex Ltd., Aschaffenburg, Germany) and an analytical column (Phenomenex Chirex 3126 (D)-penicillamine 150 × 4.6 mm: Phenomenex Ltd., Aschaffenburg, Germany) as described by Kroger et al. [30]. In total, 500 mg of sample was homogenized for 10 min with 1 mL of a copper (II) sulphate solution. Subsequently, 50 μL of Carrez I and II solutions was added followed by a further homogenization. Moreover, the samples were centrifuged (10 min at 14,800× *g*); the supernatant was filtered through a cellulose–acetate filter. The analysis was performed on aliquots (400 μL) of the filtrate mixed with 600 μL of copper (II) sulphate (0.5 mmol/L).

Berthelot reaction was applied to assess the ammonia content. The samples were defrosted on ice and 100 mM 3-(N-morpholino) propanesulfonic acid (pH 7) was used to dilute (1:2) the samples. Subsequently, the samples were homogenized for 1 min and incubated on ice for 10 min. Finally, the samples were centrifuged (17,000× *g* for 10 min at 4 °C), the pH of the supernatant was corrected to 7, and extinction was determined at 620 nm in a Tecan Sunrise microplate reader (Tecan Austria GmbH, Grödig, Austria).

### 2.4. Statistical Analysis

All parameters concerning the microbial count were assessed using the non-parametric Kruskal–Wallis test due to the low number of animals involved in the trial. The mean value and standard error were reported. The results obtained from the in vitro trial were analysed using two-way ANOVA, with the substrates and inoculum as a fixed factor. The level of significance was α = 0.05. The HSD Tukey Post-hoc test was used when the level of significance was less than 0.05. The Shapiro–Wilk test was used to ascertain the normal distribution of the data. All analyses were performed using JMP, 14 (JMP^®^, Version 14 SW, SAS Institute Inc., Cary, NC, USA, 1989–2019).

## 3. Results

### 3.1. In Vitro Fermentation

The faeces of dogs fed the HL1 diet had a higher organic matter degradability (OMD) compared to the HL2 and HS diets (*p* < 0.01) (Table 4). It showed the lowest level of gas production after 24 h of incubation (*p* < 0.01). Comparing the substrates, MOS showed the highest value of OMD while cellulose showed the lowest (*p* < 0.01). The latter also showed the lowest volume of gas production, whereas inulin produced the highest gas production (*p* < 0.01). The interaction between the fixed factors was significant (*p* < 0.01), which could be due to the different trends registered for the same substrate when incubated with different inoculum. Otherwise, cellulose produced a limited volume of gas with all the inoculum (mean OMCV 9.87 ± 1.37 mL).

In Table 5, the end-products of the fermentation faeces are reported. The pH was significantly affected by the incubated substrates. Inulin induced the lowest (*p* < 0.01) level of pH whereas cellulose induced the highest (*p* < 0.01). Regarding the volatile fatty acids, the HL1-group resulted in the highest (*p* < 0.01) production of acetate compared to the HL2 and HS groups; the latter showed the lowest level (*p* < 0.01). Otherwise, regarding the propionate production, HS group faeces produced the highest (*p* < 0.01) amount while the HL2 group showed the lowest (*p* < 0.01) percentage. The hemp group had the lowest (*p* < 0.01) percentage of total VFA, butyrate, and valerate compared to the other groups. Tallow group faeces showed the highest levels of VFA and valerate (*p* < 0.01), whereas this group had the lowest level of iso-valerate and branched-chain fatty acid (BCFA) (*p* < 0.01, *p* < 0.05, respectively). The highest (*p* < 0.05) amount of BCFA was reported by the HS group.

Comparing substrates, cellulose had the highest (*p* < 0.01) amount of acetate, iso-butyrate, iso-valerate, valerate, and BCFA while inulin showed the lowest (*p* < 0.01) amounts. Otherwise, inulin was the highest (*p* < 0.01) for VFA, propionate, and butyrate, and cellulose showed the lowest (*p* < 0.01) amount. All the interactions were significant (*p* < 0.01) except for pH and iso-butyrate.

### 3.2. Bacterial Cell Counts

In Table 6, the bacteria cell count is reported. The HL1 group had the highest counts of *Lactobacillus* spp., Enterobacteria, and *Clostridium coccoides* cluster (*p* < 0.05). Otherwise, the faeces of the HS group had the lowest (*p* < 0.05) count of Enterobacteria and the highest (*p* < 0.05) of *Clostridium coccoides*. No significant differences were observed in the cluster comparing the diets that were richer in lipids.

Regarding the percentage of volatile fatty acids in the dogs’ faeces (Table 7), the HS groups showed the highest amount of total VFAs and propionate (*p* < 0.01) while the faeces of the HL1 group had the highest proportion of acetate and butyrate (*p* < 0.01). Total VFAs and propionate increased (*p* < 0.01) in the faeces of dogs that were fed the high-starch diet compared to both groups that were fed the high-lipid diets. The HL2 group faeces had a higher percentage of iso-butyrate, iso-valerate, valerate, and BCFA (*p* < 0.01). Similarly, the diet with the tallow lipid source resulted in a higher (*p* < 0.01) level of BCFAs.

Table 8 summarizes the other metabolites in the faeces. Both HL diets showed a lower amount of D- and L-lactate compared to HS. Comparing the HL diets, the HL2 group faeces showed the lowest level of D-lactate (*p* < 0.05), and HL1 had the lowest amount of L-lactate (*p* < 0.05). Additionally, HL1 reported the lowest level of ammonia compared to the HS and HL2 groups (*p* < 0.05).

## 4. Discussion

In the present study, two lipid sources (vegetable vs. animals) in dogs’ diets were tested. In vitro fermentation evaluation and in vivo measurements were performed to investigate the effects on faecal bacteria related to metabolic activity and population. According to the experimental scheme, a 3.5% supplement of both lipid sources was added to the same commercial diet. Some differences were observed between the HL1 and HL2 diets in terms of the percentage of ether extract and protein due to the different natures of each supplement. Indeed, hemp oil contains protein residues from the oil extraction. 

Regarding the results obtained in vitro, the substrate and inoculum effects were evaluated. Concerning the substrate effect, MOS and inulin were highly fermentable, while cellulose reported a low fermentation level. The fermentation parameter results registered for inulin and cellulose are in accordance with a previous study [34]. These results confirm the prebiotic effect of inulin [35]. Otherwise, the limited volume of gas registered during the incubation with MOS could indicate that the highest organic matter degradability might be due to filtration problems related to the specific particle dimension, as indicated by Calabrò et al. [36]. Moreover, the low production of VFAs and butyrate observed with MOS during incubation might suggest that this substrate was partially filtered rather than fermented. This is probably due to the thin granulometry of the MOS substrate. Butyrate is a VFA derived from carbohydrate and protein fermentation in the large intestine. It is considered an important energy source for the colonic epithelium and regulates cell growth and differentiation [37,38]. The low fermentability of MOS could be due to the specific fungal strain or the extraction process. Indeed, in a previous study [39], six *Saccharomyces cerevisiae* cell wall samples, derived from three different production processes, were incubated with dog faecal inoculum. The authors reported significant differences in the organic matter degradability and gas production. These differences were related to the production process and mannans and glucans concentrations. Moreover, some studies carried out in vivo in carnivores showed the different fermentability of MOS supplements [40,41,42]. 

Comparing the inoculum, the faeces obtained from dogs fed the HL1 diet showed the highest value of organic matter degradability while this group reported the lowest volume of gas; these results could be related to the specific characteristics of the HL1 diet. It could be possible that the higher amount of PUFAs in the HL1 diet affected the microbial activities. In this regard, the inclusion of hemp seed oil in the diet seems to increase the fermentation ability of the microbial population without high gas production. This result could indicate the usefulness of hemp oil supplementation in limiting the side effects of prebiotic administration observed in humans [43] and dogs [44].

A few studies in the literature reported information concerning the utilisation of different lipid sources and their effects on faecal microbial fermentation activity. However, more data was collected from ruminant species. Wang et al. [45] observed a reduction in gas production after testing the in vitro supplementation of several lipid sources (seeds of safflower, poppy, hemp, and camelina vs. coconut and linseed) in bovines’ diets. These authors indicated that the application of safflower and hemp seeds reduced the level of methanogenic bacteria. In a previous study, Vastolo et al. [46] observed a lower amount of gas production after incubating hemp co-products with buffalo ruminal inoculum. These results suggest that the fatty acid profile of hemp could affect the ruminal microbial population.

Regarding the bacterial cell evaluation, Enterobacteria, *Lactobacillus* spp., *Bifidobacterium* spp., and *Clostridium* spp. have been identified as being the most common intestine bacterial units of dogs. *Clostridium* clusters XIVa and IV (*Cl. coccoides* and *Cl. leptum* clusters) are present in high amounts in the colon [47]. Diets with different ingredients but similar chemical compositions have only limited effects on the intestinal microbiome composition in dogs [17]. Furthermore, *Lactobacillus*, *Bifidobacterium*, Enterobacter, Bacteroides, and *Clostridium* seem to be involved in dietary fat metabolism (digestion and absorption) [48]. In this regard, the lipid fraction of diets appears to affect some microbial groups. Indeed, the faeces of the HL1 group showed higher *Lactobacillus* spp. cell numbers. Murphy et al. [49] indicated that the fat content is related to a reduction in microbial population diversity, particularly with a change from Bacteroides to Firmicutes. In this regard, Jia et al. [50] reported that *Lactobacillus*, which belongs to the phylum Firmicutes, has a potential probiotic function, which might improve the immune response and modulate the intestinal ecosystem of healthy dogs. These results could be remarkable, considering that, normally, *Lactobacillus* is related to the dietary carbohydrates content [51]. Indeed, Salas-Manu et al. [52] and Coelho et al. [53] reported a reduction in *Lactobacillus* when the fat level in the diet decreased. Considering this, we did not observe an increase in the *Lactobacillus* number in the faeces of dogs fed the diet supplemented with tallow, despite the higher ether extract content of this diet. It is possible that the increased level of *Lactobacillus* observed in the faeces of dogs that received hemp seed oil may be ascribable to the specific fatty acids profile of the diet rather than to the percentage of dietary fat. Kilburn et al. [20] hypothesised that *Lactobacillus* would increase when dogs were fed a diet rich in fat, which could be attributed to the activity of a specific bile salt hydrolase (BSH) observed for this genus. The probiotic bacteria BSH activity has often been related to a lower cholesterol level [54]. In this regard, in a previous study [13], the authors reported a reduction in transaminase and cholesterol in dogs fed hemp oil extraction residue as a source of lipids. Otherwise, the HL1 group’s faeces registered the highest Enterobacteria cell numbers. Considering that the latter are proteolytic bacteria, a numerical increase in enterobacteria indicates higher microbial protein fermentation in the large intestine [30]. This result seems to be confirmed by the higher value of BCFA of HL1 compared to HS. In this respect, acetate, propionate, and butyrate are the main volatile fatty acids produced by bacteria. Heimann et al. [55] indicated a role of BCFA in glucose and lipid metabolism. Regarding the decrease in VFA in the HL1 and HL2 groups’ faeces, this effect could be due to the detrimental effect of lipids on the microbial population evidenced by Shen et al. [56] in humans and mice. Nevertheless, butyric acid resulted in a higher amount in dogs fed the hemp seed oil supplementation. As previously explained, this acid is the main source of energy for the colonocyte. On the other hand, D-L-lactate decreased in the HL1 group’s faeces. Considering the increase in *Lactobacillus*, this data is unexpected. Indeed, the increase in the lactate concentrations in the faeces of the dogs indicates a promotion of lactic acid bacteria in the intestine, demonstrating a linear increase in *Lactobacillaceae* [57]. Concerning the ammonia results, the HL1 group reported a significant decrease in the ammonia value compared with the HS and HL2 groups. Despite the high percentage of crude protein, the HL1 diet does not appear to affect ammonia excretion. However, the diet appears to have influenced the microbial population. It might also be possible to assume that these results may have been caused by the presence of secondary compounds in hemp. Indeed, *Cannabis sativa* plants and co-products contain a multitude of chemicals, including phytocannabinoids, terpenoids, flavonoids, and sterols [45,46,47,48,49,50,51,52,53,54,55,56,57,58].

The HL1 group showed the lowest percentage of VFAs and the highest amount of acetic acid in both evaluations. Specifically, acetic acid has been identified by Ruiz-Matute et al. [59] as the main fermentation product of Bifidobacteria and could be considered beneficial bacteria for the intestinal environment [57]. However, considering that *Bifidobacteriaceae* did not differ between the groups, the high percentage of acetate could be related to the increase in *Lactobacillus*.

As reported by Coelho et al. [53], dogs and human have similar microbial populations, indicating that canine species could be used as models of human disease. Indeed, recently, in humans, it was observed that different microbial profiles could produce the same metabolites [60]. This relationship seems to be confirmed by the high fermentable capacity recorded in vitro with the HL1 inoculum. Moreover, the low gas production is related to this high OMD percentage with the HL1 inoculum.

## 5. Conclusions

The obtained results might indicate that the quantity and quality of dietary fat affects microbial and population activities. Moreover, the microbial communities appeared to adapt to the quality of the fat inclusion in the diet without dysbiosis. Some relation between the in vitro fermentation evaluation and in vivo microbial population count was observed. The faeces of dogs fed the hemp seed oil diet showed a higher amount of *Lactobacillus*, improving the in vitro carbohydrate fermentation activity as indicated by the highest acetate production in both experiments. Furthermore, the largest number of *Lactobacillus* observed in the faeces of dogs fed HL1 seems to be indicative of a positive effect of these lipid sources. Further studies are necessary to evaluate whether these results are related to the hemp seed oil fatty acids profile or the secondary metabolites present in the plant and its products.

## Figures and Tables

**Table 1 animals-12-01368-t001:** Chemical composition of the tested diet (g/1000 kcal ME).

Diets	HS	HL1	HL2
ME (kcal/kg)	1194	1286	1370
CP	74.3	89.6	77.0
CF	9.29	27.4	21.8
EE	30.9	55.1	65.1
Ash	27.9	28.8	23.0
NFE	136	62.2	50.7

HS: high starch; HL1: high lipid hemp; HL2: high lipid tallow. CP: crude protein; CF: crude fibre; EE: ether extract; NFE: nitrogen-free extract.

**Table 2 animals-12-01368-t002:** Fatty acid profiles of the test diets administered to dogs (mg/1000 kcal ME).

Fatty Acids	HS	HL1	HL2
C4:0	1.72	4.35	4.18
C6:0	0.37	3.86	3.21
C8:0	0.02	0.24	0.08
C10:0	0.04	ND	0.08
C12:0	0.22	ND	0.26
C14:0	1.05	1.29	3.51
C16:0	11.4	28.8	47.0
C16:1	ND	0.35	0.32
C18:1 cis6	0.12	0.14	0.50
C18:0	4.60	25.0	31.3
C18:1 trans 11 (TVA)	0.51	3.09	2.45
C18:1 cis 9	19.0	46.3	71.1
C18:1 cis 10	0.08	0.62	0.57
C18:1 cis 11	0.008	0.02	0.04
C18:2 cis n-6 (LA)	9.82	34.0	19.8
C20:0	0.08	0.61	0.37
C18:3 n-6	0.02	0.37	0.96
C18:3 n-3 (ALA)	0.95	5.75	2.59
C20:2 n-6	0.05	0.24	0.63
C22:0	0.14	0.26	2.23
C20:3 n-6	0.03	0.06	0.07
C22:1	0.02	ND	0.18
C20:3 n-3	0.11	0.99	0.45
C20:4 n-6 (AA)	0.15	0.20	0.35
C22:2 n-6	0.19	0.13	0.58
C24:0	0.03	0.09	0.18
C20:5 n-3 (EPA)	0.008	0.29	0.09
SFA	38.8	173	103
MCT	2.37	8.45	7.81
MUFA	20.2	81.1	48.7
PUFA	11.3	27.2	17.7
n-6	10.3	23.9	15.7
n-3	1.07	3.33	2.03
PUFA/SFA	0.29	0.16	0.17
n-6/n-3	9.58	7.17	7.73
LA/ALA	10.3	5.93	7.65
AA/EPA	18.0	0.70	4.00

HS: high starch; HL1: high lipid hemp; HL2: high lipid tallow. ND: not detectable; C4:0: butyric acid; C6:0: caproic acid; C8:0: caprylic acid; C14:0: myristic acid; C16:0: palmitic acid; C18:1 cis6: petroselinic acid; C18:0: stearic acid; C18:1 trans 11: trans vaccenic acid (TVA); C18:1 cis 9: oleic acid; C18:2 cis n-6: linoleic acid (LA); C20:0: arachidic acid; C18:3 n-6: γ-linolenic acid (GLA); C18:3 n-3: α-linoleic acid (ALA); C20:2 n-6: eicosadienic acid; C22:0: behenic acid; C20:3 n-6; C20:3: n-3: dihomo γ-linolenic; C20:4 n-6: arachidonic acid(AA); C22:2 n-6: docosadienoic acid; C24:0: lignoceric acid; C20:5 n-3: eicosapentenoic (EPA); SFAs: saturated fatty acids; MCT: medium chain triglycerides (C6:0 + C8:0 + C10:0 + C12:0); MUFAs: monounsaturated fatty acids; PUFAs: polyunsaturated fatty acids; LA/ALA: linoleic acid/α-linolenic acid; AA/EPA: arachidonic acid/eicosapentaenoic acid.

**Table 3 animals-12-01368-t003:** PCR characteristics and primers used to assess bacterial cell numbers in dog faeces.

Specificity	Sequence	Name	Product. bp	A_T_ ^1^	Reference
*Lactobacillus* spp.	F:5′-AGCAGTAGGGAATCTTCCA-3′ R: 5′-CACCGCTACACATGGAG-3′	LAC-1 LAC-2	341	58	Rinttilä et al. [31]
Enterobacteria	F:5′-GTTAATACCTTTGCTCATTGA-3 R:5′-ACCAGGGTATCTAATCCTGTT-3′	Entero-F Entero-R	340	50	Malinen et al. [32]
*Bifidobacterium* spp.	F:5′-TCGCGTC(C/T)GGTGTGAAAG-3′ R:5′-CCACATCCAGC(A/G)TCCAC-3′	g-BIFID-F g-BIFID-R	243	58	Rinttilä et al. [31]
*Clostridium coccoides* cluster (Cluster XIVa)	AAATGACGGTACCTGACTAA CTTTGAGTTTCATTCTTGCGAA	g-Cocc-F g-Cocc-R	440	55	Matsuki et al. [33]
*Clostridium leptum* cluster (Cluster IV)	GCACAAGCAGTGGAGT CTTCCTCCGTTTTGTCAA	sg-Clept-F sg-Clept-R	239	55	Matsuki et al. [33]

A_T_ ^1^ = annealing temperature (°C).

**Table 4 animals-12-01368-t004:** Organic matter degradability and in vitro gas production parameters of groups’ pooled faeces after 24 h of incubation.

Items	OMD	OMCV
	%	mL/g
	*Inoculum* effect	
HS-group	57.2 ^B^	51.1 ^A^
HL1-group	60.1 ^A^	40.8 ^B^
HL2-group	58.6 ^B^	58.2 ^A^
	Substrate effect	
MOS	97.5 ^A^	24.8 ^B^
Inulin	76.4 ^B^	114 ^A^
Cellulose	2.02 ^C^	10.8 ^C^
	*Inoculum* x Substrate	
	***	***
MSE	1.64	46.1

HS: high starch; HL1: high lipid hemp; HL2: high lipid tallow. MOS: mannan-oligosaccharides; OMD: organic matter degradability; OMCV: cumulative volume related to incubated organic matter. Along the column, different capital superscript letters indicate a difference at *p* < 0.01; ***: *p* 0.01 < 0.MSE: mean square error.

**Table 5 animals-12-01368-t005:** In vitro end-products at 24 h of incubation.

Items	pH	VFA	Acetate	Propionate	Iso-Butyrate	Butyrate	Iso-Valerate	Valerate	BCFA
		mmol/gOM	% VFA						
	*Inoculum* effect								
HS-group	6.27	53.4 ^B^	45.6 ^C^	23.9 ^A^	3.45	19.6 ^A^	5.08 ^A^	1.87 ^B^	8.53 ^a^
HL1-group	6.18	43.2 ^C^	53.2 ^A^	22.6 ^B^	3.06	16.2 ^B^	4.81 ^A^	0.91 ^C^	7.88 ^ab^
HL2-group	6.26	72.3 ^A^	50.2 ^B^	16.8 ^C^	3.19	19.3 ^A^	4.15 ^B^	2.27 ^A^	7.34 ^b^
	Substrate effect								
MOS	6.42 ^B^	65.6 ^B^	49.9 ^B^	21.4 ^B^	3.11 ^B^	19.4 ^B^	4.51 ^B^	1.75 ^B^	7.62 ^B^
Inulin	5.45 ^C^	74.7 ^A^	43.9 ^C^	26.1 ^A^	1.04 ^C^	22.9 ^A^	1.47 ^C^	0.62 ^C^	2.51 ^C^
Cellulose	6.83 ^A^	28.5 ^C^	55.2 ^A^	16.7 ^C^	5.55 ^A^	12.9 ^C^	8.07 ^A^	2.69 ^A^	13.6 ^A^
	*Inoculum* x Substrate								
	NS	***	***	***	NS	***	***	***	***
MSE	0.08	12.4	4.47	1.17	0.22	5.60	0.33	0.07	0.96

HS: high starch; HL1: high lipid hemp; HL2: high lipid tallow. MOSs: mannan-oligosaccharides; VFAs: volatile fatty acids; BCFAs: branched-chain fatty acids. Along the column, different capital superscript letters indicate a difference for *p* < 0.01; lowercase superscript letters indicate a difference for *p* < 0.05. *** *p* < 0.001, NS: not significant; MSE: mean square error.

**Table 6 animals-12-01368-t006:** Faecal bacterial cell evaluation of dogs fed the administered diets (log/g sample wet weight).

Bacteria Cell Count	HS	HL1	HL2	*p*-Value	
				Between Diets	HL1 vs. HL2
*Lactobacillus* spp.	6.86 ± 0.22	7.11 ± 0.05	6.55 ± 0.14	0.0429	0.0142
*Bifidobacterium* spp.	2.54 ± 0.29	2.60 ± 0.21	2.20 ± 0.11	0.6376	0.2008
Enterobacteria	6.01 ± 0.22	6.91 ± 0.10	6.66 ± 0.13	0.0131	0.1594
*Cl. coccoides* cluster XIVa	9.95 ± 0.03	9.86 ± 0.04	9.77 ± 0.05	0.0203	0.1821
*Cl. leptum* cluster IV	8.53 ± 0.09	8.05 ± 0.22	8.57 ± 0.10	0.1874	0.1408

HS: high starch; HL1: high lipid hemp; HL2: high lipid tallow. *Cl*: *Clostridium.*

**Table 7 animals-12-01368-t007:** Volatile fatty acids (VFAs) present in the dogs’ faeces.

Items	Units	HS	HL1	HL2	*p*-Value	
					Between Diets	HL1 vs. HL2
VFA (mmol/g)		163 ± 6.52	87.2 ± 2.98	134 ± 9.97	0.0024	0.0105
acetate	% VFA	60.6 ± 0.40	61.4 ± 0.82	58.7 ± 0.53	0.0221	0.0209
propionate	% VFA	28.2 ± 0.42	25.9 ± 0.45	23.1 ± 0.50	0.0001	0.0033
iso-butyrate	% VFA	1.03 ± 0.07	1.31 ± 0.12	2.34 ± 0.06	0.0003	0.0008
butyrate	% VFA	8.11 ± 0.28	10.5 ± 0.43	9.40 ± 0.44	0.0035	0.0633
iso-valerate	% VFA	1.40 ± 0.08	1.21 ± 0.07	3.45 ± 0.14	0.0003	0.0008
valerate	% VFA	0.36 ± 0.02	0.30 ± 0.02	2.94 ± 0.11	0.0004	0.0012
BCFA	% VFA	2.07 ± 0.25	2.87 ± 0.36	5.51 ± 0.31	0.0003	0.0011

HS: high starch; HL1: high lipid hemp; HL2: high lipid tallow. BCFAs: branched-chain fatty acids; VFAs: total volatile fatty acids.

**Table 8 animals-12-01368-t008:** Contents of D-L-lactate, ammonia, and pH in the dogs’ faeces.

Items	HS	HL1	HL2	*p*-Value	
				Between Diets	HL1 vs. HL2
D-lactate (µL/g)	1.67 ± 0.58	0.25 ± 0.03	0.43 ± 0.14	0.0138	0.6242
L-lactate (µL/g)	1.77 ± 0.82	0.22 ± 0.06	0.25 ± 0.14	0.0468	0.6473
Ammonia (µmol/g)	59.9 ± 4.80	42.7 ± 2.24	59.3 ± 5.46	0.0255	0.0472
pH	6.75 ± 0.06	6.97 ± 0.04	7.11 ± 0.03	0.0008	0.0273

HS: high starch; HL1: high lipid hemp; HL2: high lipid tallow.

## Data Availability

Not applicable.

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
