# Peer review of "Evaluation of the Effect of Different Dietary Lipid Sources on Dogs’ Faecal Microbial Population and Activities"

_animals, 2022, doi:10.3390/ani12111368_

Round 1
Reviewer 1 Report
Comments to authors:
General comments:
Vastolo Alessandro and co-authors performed an interesting study aiming to evaluate the effects of different lipid sources (from animal and vegetal origin) on selected faecal bacterial populations and metabolites, using both in vitro and in vivo methods. However, some points need to be reviewed carefully, before the study can be considered for publication.
There are some typos and grammatical errors that should be corrected. Moreover, the standard of English needs to be improved throughout, and the manuscript would benefit from a thorough proof read by a native English speaker.
Specific comments:
Line 17: the sentence is uncompleted.
Line 18: the use of punctuation is not correct, therefore the sense of the sentence is not clear.
Lines 34 and 35: I suppose you referred to the in vivo/in vitro methods when you say both evaluations, but in the abstract you had never referred in detail to the in vivo trial, therefore this should be improved.
M&M- diets: it is not clear to me if the same canned diet was used for HL1 and HL2 diets. If so, why did you not consider to obtain iso-lipidic experimental diets? If you aimed to evaluate the effects of the two different lipid sources, a similar lipid content should have been considered. Instead, HL1 had 10g/Mcal of lipid less than HL2: what do you think about this difference in lipids content?
In vitro study: I have a concern regarding the absence of a ctrl group – meaning vessels with fecal inocula without different substrates, only the diets. In this case it is not possible to authors to distinguish between the dietary-effect from the substrate-effect. How did the authors select the substrates? Are it have been added to the different diets fed to animals? Since the authors did not gave those informations, I could not entirely had understood the comparison between results from the in vivo and in vitro studies the authors discussed later on.
In vivo study: Were the fecal samples pooled between the 2 different times of collection? If so, why did the authors choose to sampled twice at very different time points? Did you think it might have affected the dietary adaptation period?
Table 5: I think VFA instead of SCFA should be more appropriate, since the authors have put the results of lactate concentrations in a different table.
Tables 6-7-8: please align all the tables with the same layout when you refer to multiple comparison. Why did the authors omit the post-hoc results from the overall comparisons?
Lines 267-268: This sentence does not make sense to me. What does “the latter” refers to?
Discussion:
Lines 298-303 the authors did not measure the fecal lipid content and the fecal fatty acids profile in the different groups. Instead, they referred to effects exerted by the dietary lipid sources, or specific fatty acids such as MCTs, on fecal microbiota and metabolites, without knowing the lipid content of feces, or if the selected fatty acids had reached the colon or had being absorbed earlier in the GI tract (as far as I know, the digestibility If those analyses cannot be performed, it should be stated as a limit of the study. Moreover, in the introduction you’ve referred
Lastly, I have a comment about the MCT content of diets: you might specifically refer to C6 and C8 for HL1, because C10 and C12 were not detectable in this diet (table 2).
Lines 304-311: the authors referred to studies evaluating the effects of lipid sources in ruminants, while at least one study already exists in dogs that can be cited: DOI: 10.3389/fmicb.2020.564160
Please, consider to enhance the discussion part.
Author Response
There are some typos and grammatical errors that should be corrected. Moreover, the standard of English needs to be improved throughout, and the manuscript would benefit from a thorough proof read by a native English speaker.
Thank you for the observations, the manuscript has been revised in order to correct all typos and grammatical errors, and improve the English.
Specific comments:
Line 17: the sentence is uncompleted.
The sentence was removed.
Line 18: the use of punctuation is not correct; therefore the sense of the sentence is not clear.
The paragraph re-written.
Lines 34 and 35: I suppose you referred to the in vivo/in vitro methods when you say both evaluations, but in the abstract, you had never referred in detail to the in vivo trial, therefore this should be improved.
Thank you for the observation, the trials were better specified in the sentence
M&M- diets: it is not clear to me if the same canned diet was used for HL1 and HL2 diets. If so, why did you not consider obtaining iso-lipidic experimental diets? If you aimed to evaluate the effects of the two different lipid sources, a similar lipid content should have been considered. Instead, HL1 had 10g/Mcal of lipid less than HL2: what do you think about this difference in lipids content
The authors agree with the reviewer observation but in the experimental design was preferred to use the same amount of supplementation (3.5 % a.f.) to the same canned diet. Taking in count reviewer observation in discussion paragraph the authors add some consideration related to the differences between the diet in term of lipid and protein content.
In vitro study: I have a concern regarding the absence of a ctrl group – meaning vessels with fecal inocula without different substrates, only the diets. In this case it is not possible to authors to distinguish between the dietary-effect from the substrate-effect. How did the authors select the substrates? Are it have been added to the different diets fed to animals? Since the authors did not gave those informations, I could not entirely had understood the comparison between results from the in vivo and in vitro studies the authors discussed later on.
Thank for the observation, we considered the diet HS as the control one because it is usually administered to the dogs in the kennel. We add this information in the description of the experimental design and at the beginning of the description of in vitro trial. Moreover, the substrates were chosen for their fermentation characteristics in order to evaluate if the faecal inoculum differ in fermentative activities. In particular, cellulose is an unfermentable carbohydrates, while MOS and inulin were notoriously high fermentable substrates.
In vivo study: Were the fecal samples pooled between the 2 different times of collection? If so, why did the authors choose to sampled twice at very different time points? Did you think it might have affected the dietary adaptation period?
Thank you for the observation, the first sentence contains a typo, actually, we did not analyse faecal samples at 15 days
Table 5: I think VFA instead of SCFA should be more appropriate, since the authors have put the results of lactate concentrations in a different table.
The SCFA has been modified in VFA.
Tables 6-7-8: please align all the tables with the same layout when you refer to multiple comparison. Why did the authors omit the post-hoc results from the overall comparisons?
Thank you, the authors aligned all tables ‘layout. Furthermore, the authors carried out post-hoc tests only for parameters referring to in vitro evaluations, where a higher number of replications were present. Non-parametric tests were carried out for the in vivo evaluation due to the low number of dogs
Lines 267-268: This sentence does not make sense to me. What does “the latter” refers to?
Thank you, the lines have been changed.
Discussion:
Lines 298-303 the authors did not measure the fecal lipid content and the fecal fatty acids profile in the different groups. Instead, they referred to effects exerted by the dietary lipid sources, or specific fatty acids such as MCTs, on fecal microbiota and metabolites, without knowing the lipid content of feces, or if the selected fatty acids had reached the colon or had being absorbed earlier in the GI tract (as far as I know, the digestibility If those analyses cannot be performed, it should be stated as a limit of the study. Moreover, in the introduction you’ve referred
Thank you for the observation, the sentences referred to MCTs have been modified considering the lipid content in the faeces was not analyse.
Lastly, I have a comment about the MCT content of diets: you might specifically refer to C6 and C8 for HL1, because C10 and C12 were not detectable in this diet (table 2).
Thank you, that’s true C10 and C12 were not detectable. Despite this result, the MCT resulted in higher value in H-diet than in T-diet. This last suggested higher presence of MCT in hemp co-product
Lines 304-311: the authors referred to studies evaluating the effects of lipid sources in ruminants, while at least one study already exists in dogs that can be cited: DOI: 10.3389/fmicb.2020.564160
Thank you for the suggestion, the article has been useful. A sentence was added taking in count the suggested reference
Please, consider to enhance the discussion part.
The authors enhanced discussion paragraph
Reviewer 2 Report
Summary
The study investigates the effect of fat sources (plant or animal-source) on canine fecal microbiota compared to a high-starch control.
Major concerns
Methods
- Were baseline fecal samples taken before and/or after the 10-day adaptation period? No baseline sample is a design flaw.
- Were all samples within treatment pooled for in vitro gas production? If so, this reduces your sample size to 1 for each inoculum. Please clarify.
- It was stated that fecal samples were collected at day 15 and 30 post-treatment, but it does not appear that the 15-day results are presented?
- Line 324 – referring to numerical differences is inappropriate if not statistically significant.
- Line 332 – butyrate is not the primary energy source for enterocytes. Glutamine is the primary energy source for enterocytes. Butyrate is an energy source for colonocytes. This is an incorrect statement.
Minor Concerns
- English grammatical errors need revision throughout
- Line 292 – random close parentheses.
Author Response
Major concerns
Methods
Were baseline fecal samples taken before and/or after the 10-day adaptation period? No baseline sample is a design flaw.
Faecal baseline samples were not collected because each parameter was compared with control group (HS).
Were all samples within treatment pooled for in vitro gas production? If so, this reduces your sample size to 1 for each inoculum. Please clarify.
Thank you for the observation. a faecal pool of each group was prepared to limit individual effect. The authors incubated 4 flasks for each inoculum with each substrate and we considered the flask as a replication.
It was stated that faecal samples were collected at day 15 and 30 post-treatment, but it does not appear that the 15-day results are presented?
Thank the first sentence contains a typo we had not analysed faecal sample of 15 day and consequently we removed the typo
Line 324 – referring to numerical differences is inappropriate if not statistically significant.
The sentence was removed
Line 332 – butyrate is not the primary energy source for enterocytes. Glutamine is the primary energy source for enterocytes. Butyrate is an energy source for colonocytes. This is an incorrect statement.
Thank you, the sentence was changed
Minor Concerns
English grammatical errors need revision throughout
Thank you, the English has been revised throughout the manuscript
Line 292 – random close parentheses.
The error has been corrected
Reviewer 3 Report
See the file

Author Response
Line 11: no “.” Before while in “body temperature. While”
Thank you, the sentence has been corrected
Line 14: please add the lipid contents in the sentence
The lipid contents have been added to the sentece
Line 18: add a “,” after supplementation
Done
Line 22: add a “.” at the end of the sentence
Done
Lines 48-50: please add a reference on MCT effect on epilepsy in dogs
The reference was added
Lines 72-76: the sentence is not clear at all. I suggest “However, despite the authorization of the European Food Safety Authority (EFSA) to use hemp seeds and hemp seed co-products as ingredients in all animal species [13], very few data have been published in the literature concerning the use of hemp and its co-products as a source of PUFAs in pet food.”
Thank you for the suggestion, the lines were modified
Lines 76-79: here also, please explain simply the high variability of microbiote in dogs, and its variation with age, diet…. “The gut microbial population 76 is fundamental from first stage of life to support intestinal development. Several studies 77 have indicated a high individual variability [14,15]. Several factors could affect the micro- 78 bial population in the gut such as age, environment, and dietary composition [16].”
Thank you for the suggestion, the senteces were modified
Line 80 “it seems evident that nutrition represents one of the most important factors”….no, it could be . Replace with “nutrition represents one of the …”
The authors replaced the sentences as suggested by the reviewer
Lines 81-83: “Consequently, in the last years, researchers were stimulated to evaluate the effect of different nutrients [17] or diet physical forms [18-14]on microbial population.” No, it is not the consequence, it is thanks to this work that the influence of diet on the microbiota has been highlighted.
Thank you, the authors revised the sentences
Please write [14-18] and add a space after the parenthesis
The errors were corrected
Lines 83-84: “Numerous studies described the positive effect of carbohydrates sources on microbiota composition.” You must cite several studies and detail which carbohydrates have an effect and how it is a positive effect
The studies have been added to references
Line 87: No “:” after purpose
Done
Lines 103-107: “A canned commercial diet was supplemented with the same amount (3.5 % as fed) of swine tallow or hempseed cake. The obtained diets, named High Lipid 1 (HL1) and High Lipid 2 (HL2) were compared with a canned diet, usually utilized into the kennel, richer in starch (HS). Each diet was randomly assigned to one group and daily administered in a ratio of 110 kcal/kg0.75 of metabolizable energy (ME) [19].” I don’t understand why you did not try to obtain three isoenergetic regimen, or the same quantities of CP ….Please explain your choice.
The authors agree with the reviewer observation but in the experimental design was preferred to use the same amount of supplementation (3.5 % a.f.) to the same canned diet.
Table 1: it is impossible that the ME contain of HS regimen was 119 kcal/g (and the other also). It is a mistake, please correct it.
The mistake has been corrected
Lines 152, 168, 281, 306, 309, 320, 330, 346 and in table 3: et al. in italics (?)
Corrected
Line 163: no “.” after “numbers” in “The analysis of bacterial cell numbers. D-L lactate and ammonia have been performed at the Institute of Ani- 163 mal Nutrition, Freie Universität Berlin.”
The line has been changed
Line 237: “except 237 for pH and butyrate”, no for iso-butyrate in the table
corrected
Line 251: under table 6, please add Cl. as an abbreviation of Clostridium.
Clostridium has been added
Lines 267-268: “Additionally, the latter reported the lowest level of ammonia compared 267 to HS and HL1 groups (p<0.05). It is a mistake, it is HL2 group, please correct.
The sentence has been modified
Lines 289-291: “. Indeed, incubating six Saccharomyces cerevisiae cell wall samples, derived from three different production processes, by in vitro gas production technique using dog faeces as inoculum”. I don’t understand this sentence.
The sentences have been clarified
Line 292: remove + and the parenthesis after glucans
The plus and parenthesis have been removed
Line 297: no differences of what ? were observed between HS and HL2 diets.
The line has been changed
Line 301: high or higher ?
The line was corrected, the right word was higher
Line 315: “Clostridium coccoides cluster and C. leptum cluste”. Please use Cl. or Clostridium.
Modified
Lines 315-317: “In dogs, diets with different ingredients but similar chemical compositions can have only limited effects on the composition of the intestinal microbiome [16].” Please replace by: In dogs, diets with different ingredients, but similar chemical compositions had only limited effects on the composition of the intestinal microbiota
The authors replace the sentences as suggested by the reviwer
Line 319: faeces or feces, not faces
The line was changed
Line 325: Cl. leptum, not Cl. Leptum
corrected
Line 333: D-L-lactate or DL-lactate
modified
Round 2
Reviewer 1 Report
I would like to thank the Authors for improving the manuscript. However, this comment from the previous report is still pending:
reviewer -->M&M- diets: If you aimed to evaluate the effects of the two different lipid sources, a similar lipid content should have been considered. Instead, HL1 had 10g/Mcal of lipid less than HL2: what do you think about this difference in lipids content?
authors--> The authors agree with the reviewer observation but in the experimental design was preferred to use the same amount of supplementation (3.5 % a.f.) to the same canned diet. Taking in count reviewer observation in discussion paragraph the authors add some consideration related to the differences between the diet in term of lipid and protein content.
I cannot find those considerations referring to this study. How do the authors explain the possibile effects exerted by a difference in lipid content of the two diets?
Author Response
reviewer -->M&M- diets: If you aimed to evaluate the effects of the two different lipid sources, a similar lipid content should have been considered. Instead, HL1 had 10g/Mcal of lipid less than HL2: what do you think about this difference in lipids content?
The differences in lipid content into the diet could be due to the different nature of supplements. Indeed the hemp oil might still contain residues of protein, for this reason, it could influenced the diet differently than the tallow.
authors--> The authors agree with the reviewer observation but in the experimental design was preferred to use the same amount of supplementation (3.5 % a.f.) to the same canned diet. Taking in count reviewer observation in discussion paragraph the authors add some consideration related to the differences between the diet in term of lipid and protein content.
I cannot find those considerations referring to this study. How do the authors explain the possibile effects exerted by a difference in lipid content of the two diets?
Thanks for the observations. The authors improved the discussion section regarding these aspects
Round 3
Reviewer 1 Report
The authors have addressed all the comments from my side.